# Regulation of Carotenoid Biosynthesis and Degradation in Lettuce (*Lactuca sativa* L.) from Seedlings to Harvest

**DOI:** 10.3390/ijms241210310

**Published:** 2023-06-18

**Authors:** Galina Brychkova, Cleiton Lourenço de Oliveira, Luiz Antonio Augusto Gomes, Matheus de Souza Gomes, Antoine Fort, Alberto Abrantes Esteves-Ferreira, Ronan Sulpice, Peter C. McKeown, Charles Spillane

**Affiliations:** 1Genetics & Biotechnology Laboratory, Agriculture, Food Systems & Bioeconomy Research Centre, Ryan Institute, School of Biological & Chemical Sciences, University of Galway, University Road, H91 REW4 Galway, Ireland; cleiton.oliveira@ufla.br (C.L.d.O.);; 2Department of Agriculture, Federal University of Lavras (DAG/ESAL), Aquenta Sol, Lavras 37200-000, MG, Brazil; 3Departamento de Agricultura, Universidade Federal de Lavras (UFLA), Lavras 37203-202, MG, Brazil; 4Laboratory of Bioinformatics and Molecular Analysis, Institute of Genetics and Biochemistry, Campus Patos de Minas, Federal University of Uberlandia, Av. Getúlio Vargas, 230, Patos de Minas 38700-103, MG, Brazil; 5Department of Life & Physical Science, Technological University of the Shannon: Midlands Midwest, N37 HD68 Athlone, Ireland; 6Plant Systems Biology Laboratory, Agriculture, Food Systems & Bioeconomy Research Centre, Ryan Institute, School of Biological & Chemical Sciences, University of Galway, University Road, H91 REW4 Galway, Ireland

**Keywords:** carotenoids, carotenoid synthesis, gene expression, *Lactuca sativa*, lettuce development, biomarkers, biofortification

## Abstract

Lettuce (*Lactuca sativa* L.) is one of the commercially important leafy vegetables worldwide. However, lettuce cultivars vary widely in their carotenoid concentrations at the time of harvest. While the carotenoid content of lettuce can depend on transcript levels of key biosynthetic enzymes, genes that can act as biomarkers for carotenoid accumulation at early stages of plant growth have not been identified. Transcriptomic and metabolomic analysis was performed on the inner and outer leaves of the six cultivars at different developmental stages to identify gene-to-metabolite networks affecting the accumulation of two key carotenoids, β-carotene and lutein. Statistical analysis, including principal component analysis, was used to better understand variations in carotenoid concentration between leaf age and cultivars. Our results demonstrate that key enzymes of carotenoid biosynthesis pathway can alter lutein and β-carotene biosynthesis across commercial cultivars. To ensure high carotenoids content in leaves, the metabolites sink from β-carotene and lutein to zeaxanthin, and subsequently, abscisic acid needs to be regulated. Based on 2–3-fold carotenoids increase at 40 days after sowing (DAS) as compared to the seedling stage, and 1.5–2-fold decline at commercial stage (60 DAS) compared to the 40 DAS stage, we conclude that the value of lettuce for human nutrition would be improved by use of less mature plants, as the widely-used commercial stage is already at plant senescence stage where carotenoids and other essential metabolites are undergoing degradation.

## 1. Introduction

An important objective of sustainable crop production is to increase the production of crops that offer enhanced nutritional values. Consumption of bioactive compounds such as carotenoids, especially β-carotene and lutein, is linked to improvements in human health and a reduced incidence of non-communicable diseases [1,2]. In animals, carotenoids act as antioxidants and prevent oxidative damage to cells [3,4]. Increased consumption of pro-vitamin β-carotene enriched food reduces Vitamin A deficiency [5,6], which remains one of the major health non-communicative problems in developing countries: those affected suffer increased disease susceptibility, ocular degeneration and even permanent blindness [7,8]. Lutein-enriched nutrition can also prevent eye diseases, improve photoprotection and reduce the risks of age-related macular degeneration by 80% [9,10].

The shift toward a vegetable-based diet also requires a better understanding of carotenoid accumulation in plants as, for the vegan population, plants are the only source of vitamin A precursors [5]. Therefore, understanding the relationship between the expression levels of carotenoid biosynthesis genes and final levels of β-carotene and lutein accumulation is important for plant breeding and for metabolic engineering approaches for increasing delivery of these health-promoting metabolites. There is a particular interest in identifying which of the genes involved in carotenoids biosynthesis are expressed early in plant development. Early gene expression profile could allow a robust prediction of carotenoids levels at the time of harvest and greatly accelerate screening for improved germplasm. Several studies have compared gene expression levels with carotenoid accumulation across plant life cycles, e.g., apple [11], tomato [12], cabbage [13], squash [14], carrot [15], chrysanthemum [16] and kiwifruit [17]. However, the networks underlying gene-to-metabolite are largely unknown, and the relationship between expression levels of carotenoid pathway genes and carotenoid levels in green leafy vegetables is not yet established.

In most plants, carotenoids usually accumulate in specialized plastids (chromoplasts) in organs such as roots, tubers, flowers and fruits and cause characteristic orange or yellowish coloration [18]. In photosynthetic tissues, carotenoid accumulation occurs in the chloroplasts, where they act as accessory pigments in light harvesting and protect against photooxidative damage [19,20]. In fact, lutein and β-carotene levels in green leafy vegetables such as lettuce, kale and sweet basil have been shown to significantly correlate with chlorophyll concentration and resulting green color intensity [21,22,23].

More than 700 types of carotenoids have been identified in photosynthetic organisms [4]. Carotenoids are involved in light harvesting and, as such, contribute to effective photosynthesis; they are also involved in non-photochemical quenching and protecting plant tissues from excess light or heat stress. In higher plants, carotenoids are synthesized through mevalonate pathway from Acetyl-Co-A in cytosols and non-mevalonate pathways that starts from pyruvate in chloroplasts. While these pathways are highly conserved, the regulation of carotenoid biosynthesis is tissue- and species-specific [2]. Both pathways result in C5-isopentenyl pyrophosphate (IPP), the starting point of carotenoids biosynthesis. The four molecules of IPP undergo condensation to form a single C20-geranylgeranyl diphosphate (GGPP) molecule [2,18,24]. GGPP is converted to phytoene by the phytoene synthase (PSY) and subsequently converted to lycopene via 9,15,9′-tris-cis-ζ-carotene by phytoene desaturase (PDS) and ζ-carotene desaturase (ZDS) [2]. The ends of the linear carotenoid lycopene can then be cyclized by lycopene β-cyclase (LCY-β1) and/or lycopene ε-cyclase (LCY-ε) and subsequently modified by hydroxylation [25]. The combined action of LCY-β1 and LCY-ε allows synthesis of α-carotene from lycopene, while LCY-β alone leads to β-carotene. After synthesis, α-carotene can be hydroxylated to form lutein by either α-ring or β-ring hydroxylase (CHY α, β); β-carotene can also be hydroxylated to zeaxanthin by CHYβ. Zeaxanthin is then transformed into violaxanthin via antheraxanthin by zeaxanthin epoxidase (ZEP) [2,26] (Appendix A). In addition to the usual ε-β and β-β branches in lettuce and some other plants, the third ε-ε-branch from lycopene is identified (Appendix A) [27,28]. This branch is activated by LCYE in the absence of active LCYB, and a number of unusual carotenes are synthesized from ε-carotene, including delta-carotene, epsilon-carotene and lactucaxanthin (epsilon,epsilon-carotene-3,3’-diol).

A range of molecular mechanisms regulate carotenoid biosynthesis and accumulation in plastids during plant development. Deoxyxylulose 5-phosphate synthase (DXS) and deoxyxylulose 5-phosphate reductoisomerase (DXR) catalyze the first two enzymatic steps of the methylerythritol 4-phosphate (MEP) pathway to supply the isoprene building-blocks of carotenoids and may be rate-limiting in different species. Indeed, the overexpression of DXS and DXR increases carotenoid and chlorophyll leaf content in *Daucus* and *Nicotiana* but has no affect in, e.g., *Lavandula* [29]. Differences in the transcript levels of the genes involved in carotenoid metabolism can be directly related to the level of carotenoid synthesis or degradation in many species, suggesting key roles for biosynthetic control at the transcriptional level [29,30]. For example, carotenoid accumulation in harvested ripe apples can be predicted during fruit development from increased levels of carotenoid isomerase (*CRTISO*) and *LCY-ε* transcripts [11]. Similar predictive relationships have been found in *Daucus* (carrot), in which increased transcript levels of *LCY-ε* could be correlated with final lutein concentration, while *ζ-carotene desaturase (ZDS)*, *ZDS1* and/or *ZDS2* transcript levels could be correlated with the final lycopene concentration [15].

The presence of chlorophyll-binding proteins and lipoproteins that sequester carotenoids in plastids can affect carotenoid accumulation [31]. Source–sink relationships affecting fluxes of metabolites through biosynthetic pathways also affect carotenoid accumulation in plants [32]. In fruits, the conversion of chloroplasts to chromoplasts during fruit development makes chromoplasts the major storage structures of carotenoids metabolites in ripe fruits [33]. However, in green leafy vegetables, such conversions do not occur, and such plants form carotenoid crystals to increase accumulation capacity [34,35]. A similar mechanism has been identified in cauliflower in which plastid differentiation is affected by a mutant gene *Or*, which causes β-carotene accumulation in plant curds and increases the sink capacity [36]. In addition, the degradation of carotenoids produced at the end of the pathway can affect carotenoid concentrations [37].

Lettuce (*Lactuca sativa* L., Asteraceae) is the most commercially important and widely consumed green salad crop in the world and has significant potential to provide carotenoids and other nutrients for human health [5]. However, lettuce cultivars display large variations in their nutritional composition. For example, the most widely consumed lettuce, the iceberg type, shows the least amount of antioxidant activity of the main lettuce cultivars [23,38,39]. Despite the importance of carotenoids for human health, the homeostasis of carotenoids during the lettuce leaf development remains surprisingly unexplored.

The aim of our research was to understand if carotenoids accumulation was differently regulated in plant with distinct architecture and leaf color. In this study, our objective was to investigate the role of 17 genes involved in carotenoid metabolism in lettuce leaves and to determine their expression in six commercial cultivars at different plant developmental stages. We studied differential gene expression changes and compared them to carotenoids (β-carotene and lutein) and chlorophyll metabolic changes in the inner and outer leaves throughout the plant life cycle. Through a gene-to-metabolic network analysis, we then tested the role of carotenoid biosynthetic and degradation pathways to understand β-carotene and lutein accumulation in lettuce plants for further use in lettuce breeding programs.

## 2. Results

### 2.1. Growth and Carotenoid Content Analysis of Commercial Lettuce Cultivars during Plant Development

The development of six contrasting commercial lettuce cultivars was followed from seedling stage (20 DAS) through mature stage (40 DAS) until commercial stage (Comm, 60 DAS)—that is, the usual stage at which lettuce is harvested for consumption. All cultivars had different growth rates, and the biggest difference between cultivars was already evident at 40 DAS, where Veronica and Grand Rapid cultivars had at least two-fold larger leaf areas of the outer leaves compared to Dark Land and Dragoon varieties, while Salinas 88 had the largest inner leaves (Figure 1).

As expected, the biggest difference between the six cultivars was visible at the commercial stage, when plants displayed extreme variation in leaf shape, thickness, leaf size and leaf color. At the commercial stage, leaf color ranged from light green for Grand Rapid and Veronica to dark green for Salinas 88 and Parris Island or dark green/purple for Dragoon and Dark Land (Figure 1).

We quantified the β-carotene, lutein and chlorophyll a, chlorophyll b and total chlorophyll concentrations at seedling, 40 DAS and Comm stages in the inner leaves and at 40 DAS and Comm stage in the outer leaves of six commercial lettuce cultivars (Figure 2). All lettuce cultivars exhibited the lowest concentrations of β-carotene and lutein at the seedling stage (Figure 2A,B). The concentration of lutein (Figure 2A) had nearly doubled by 40 DAS in all cultivars, except in Veronica, which had the lowest lutein content (25 μg g^−^^1^ FW). In the inner leaves at Comm stage, the lutein content was either reduced (e.g., Dark Land and Dragoon) or remained unchanged from 40 DAS (e.g., Grand Rapid and Veronica) (Figure 2A). At the same time, the lutein content in the outer leaves was slightly increased at commercial stage comparing to 40 DAS.

As with lutein, the β-carotene level was the lowest at the seedling stage (Figure 2A,B). However, β-carotene displayed the highest variability between the cultivars and leaves. For example, in the inner leaves of Dark Land β-carotene, content increased three-fold at 40 DAS (288.6 ± 19.7 μg g^−^^1^ FW), and then at 60 DAS reduced to 110.4 ± 14.5 μg g^−^^1^ FW, only slightly higher than at 20 DAS (65.2 ± 9.6 μg g^−^^1^ FW). The β-carotene content in the outer leaves in Dark Land also reduced by 50% between 40 DAS and Comm stage. Interestingly, the inner leaves of Dark Land had higher β-carotene levels at 40 DAS compared to the outer leaves. In Veronica’s inner leaves, β-carotene level was just slightly increased from 42.5 ± 6.4 μg g^−^^1^ FW to 59.1 ± 3.85 μg g^−^^1^ FW during the first 20 days of maturation, and β-carotene content had not changed by Comm stage.

Grand Rapid and Salinas88 had 1.5–2-fold increase in β-carotene level at 40 DAS for inner leaves (Figure 2B). We observed that dark-leaved varieties had significant reductions in the β-carotene level (Figure 2B) at commercial stage in both inner and outer leaves (like Dark Land, Dragoon, Parris Island), while light-green cultivars Grand Rapid and Veronica had much smaller reductions in β-carotene compared to at 40 DAS. However, the overall β-carotene was the smallest among all cultivars at all developmental stages tested. These findings are critical for lettuce commercial stage definition and further lettuce selection if plants are to be a key source of provitamin A for human consumption [5].

Analysis of Chlorophyll a, b and total chlorophyll (Figure 2C–E) revealed that dark-leaved varieties consistently had the highest chlorophyll levels in the inner leaves up to 40 DAS, with significant reduction of chlorophyll content in the inner leaves at Comm stage in all except Parris Island, likely due to plant architecture. In contrast, chlorophyll in outer leaves remained steadily high or even increased slightly (1.3× in Dragoon) up to 60 DAS. Light-green cultivars, as expected, had the lowest chlorophyll content in the leaves, and chlorophyll content was significantly reduced by the Comm stage. In all genotypes, the concentration of chlorophyll a was higher than chlorophyll b at all leaf ages (Figure 2C,D). Only in Grand Rapids was total chlorophyll higher in inner than outer leaves (Figure 2E). The Pearson correlation between the normalized metabolites (Figure 3A) revealed very strong correlation between lutein and β-carotene concentrations (R^2^ = 0.86, *p* < 0.001), with two distinct clusters present—cluster 1 for lutein, β-carotene and total carotenoids and cluster 2 for chlorophyll a, b and total chlorophyll (Figure 3A). The β-carotene had negative correlation with chlorophylls in leaves, while lutein concentration in lettuce had weak relationship with leaf color (R *^2^*= −0.53, *p* > 0.05, Figure 3A). Regression analysis confirmed that lutein concentration in leaves could predict β-carotene content, as suggested by the Pearson correlation (R^2^ = 0.497, *p* < 0.001, Figure 3B, Appendix A), while chlorophyll a, b or total chlorophyll levels have no direct effect on β-carotene content in leaves (*p* > 0.05, Appendix A) if the model was excluding the plant developmental stage, leaf’s location or variety. However, the full model, accounting for effect of variety, lettuce developmental stage and location of leaves used for the samples, had a better correlation coefficient (R^2^ = 0.77, *p* < 0.001) and illuminated the importance of the Chlb (*p* = 0.01) and total chlorophyll as a predictor of β-carotene content in lettuce.

### 2.2. Relative Expression of Carotenoid Biosynthetic Pathway Genes during Lettuce Plant Development

To determine whether the expression level of regulatory carotenoid biosynthetic genes affects carotenoid accumulation [2,24] in different lettuce cultivars, we searched for candidate carotenoid biosynthetic genes in the lettuce genome project sequence (https://lgr.genomecenter.ucdavis.edu, last accessed on 17 June 2023). We identified 17 candidate genes (Appendix A) encoding key enzymes of the core biosynthesis of carotenoids. In several cases, these genes were present as low copy number families: for example, only two copies of geranylgeranyl diphosphate (*GGPS* and *GGPPS*), phytoene synthase (*PSY1* and *PSY2*) and lycopene β-cyclase (*LCY-β1* and *LCY-β2*) were identified.

The expression patterns of different genes expression changed during plant development (Figure 4 and Appendix A).

The heat maps generated from the gene analysis expression analysis revealed that transcription of the carotenoid genes is modulated in a largely uniform manner in response to developmental stages but did vary between leaf position (Figure 4). In general, the expression of most (13/17) carotenoid biosynthesis genes declines progressively throughout plants development. The highest fold change of expression levels between SDL and 40 DAS stages was noticed for *DXS* (358 ± 101-folds decline, depending on genotype, Appendix A); *LCY-B2* (110 ± 24 folds decline, depending on genotype, Appendix A); *LCY-ε* (232 ± 49 folds decline); and *CHY-β* (170 ± 54 folds decline, depending on genotype, Appendix A). Other genes involved in carotenoids biosynthesis, such as IPI, declined circa 2.4-fold, comparing expression at SDL to 40 DAS stages in the inner leaves, and the highest decline was noticed at the Comm stage (10.4 ± 2.7 folds decline, depending on genotype, Appendix A). The four exceptions are ε-ring hydroxylase (*CHY-ε*), phytoene desaturase (*PDS*), ζ-carotene desaturase (*ZDS*) and phytoene synthase (*PSY1*), whose expression levels remained the same at 40 DAS and were reduced only at the Comm stage (Appendix A). Notably, some genes had different patterns between the inner and outer leaves. For example, geranylgeranyl diphosphate (*GGPPS*), ε-ring hydroxylase (*CHY-ε*) and phytoene synthase (*PSY1*) genes were higher at the Comm stage than at 40 DAS in the outer leaves, unlike the inner leaves, where these genes were down-regulated in all cultivars (Figure 4).

To obtain a global perspective on the expression pattern of the gene involved in lutein, β-carotene and chlorophyll biosynthesis during plants development, we performed principal component analysis (PCA) on their normalized expression levels (Figure 5). The first three principal components (PCs) accounted for 75.71% of the total variance among the samples (Figure 5, left and right). Time of sampling had the highest effect on the expression profile in the inner and outer leaves. The strongest correlation was driven by the age of plants and sample source (inner or outer leaves).

We further analyzed the patterns of gene expression at each developmental stage (Appendix A). Some interesting patterns were observed. For example, CRTISO, a key enzyme encoding lycopene synthesis, was co-expressed with many early genes in the carotenoids pathway when all samples were analyzed across all developmental stage, but when analyzed at the early developmental stage, a significant positive correlation between *CRTISTO* was observed only with the genes responsible for zeaxanthin degradation, *VDE* and *ZEP* (r = 0.89 and r = 0.90, respectively). Genes involved in β-carotene biosynthesis, *LCYβ1*, and *LCYβ2*, were co-expressed with genes involved in β-carotene conversion into abscisic acid. In addition, *LCYβ1* expression had significant positive correlation to *LCY-ε* (r = 0.87, *p* < 0.05), *CHY-ε* (r = 0.95, *p* < 0.05), *CHY-β* (r = 0.94, *p* < 0.05) and *ZEP* (r = 0.83, *p* < 0.05) in the inner leaves at the commercial stage, while in the outer leaves at the same developmental stage, *LCYβ1* expression had significant positive correlation only to *VDE* gene (r = 0.91, *p* < 0.05) (Appendix A).

### 2.3. Regulation of Carotenoid Biosynthetic during Lettuce Plant Development

To elucidate the role transcriptional regulation plays in coordinating carotenoids biosynthesis, in particular lutein and β-carotene, and the synthesis of other functionally related compounds like chlorophyll a and chlorophyll b, we summarized the normalized gene expression of all lettuce cultivars and presented the metabolic map of gene expression pattern and fold changes of respective metabolites (Figure 6) and performed correlation analysis between chlorophylls and carotenoid synthesis through gene expression analysis and carotenoid and chlorophyll content (Appendix A). Deoxyxylulose 5-phosphate synthase (*DXS*) and deoxyxylulose 5-phosphate reductoisomerase (*DXR*) are the enzymes that catalyze the first steps of the pyruvate conversion to 1-deoxy-D-xylulose-5-phosphate and 2-C-methyl-D-erythriol-4-phosphate within the methylerythritol 4-phosphate (MEP) pathway to supply the isoprene building-blocks of carotenoids [29]. The transcript level of *DXS* was significantly higher in the inner leaves at seedling stage (20DAS) compared with other tissues of both stages, suggesting its relevant roles in photosynthetic young tissues. A single copy-downstream gene of *DXS*, *DXR*, was strongly expressed in the inner leaves at 20 DAS, but the highest expression level was noticed at the commercial stage, while in the outer leaves the *DXR* gene was downregulated at 40 DAS and at the commercial 60 DAS stage. The expression of genes encoding early pathway enzymes controlling the metabolite flux into the carotenoid pathway and chlorophyll synthesis, in particular isopentenyl pyrophosphate (IPP) isomerase (IPI) and geranylgeranyl pyrophosphate synthase (*GGPS*) which leads to gerangerandyl diphosphotase (GGPP) synthesis [40]) was also high at early developmental stages. Hence, the transcriptional regulation of the GGPPS gene serves as an important regulatory node in coordinating carotenoid and chlorophyll a and chlorophyll b biosynthesis.

The peak in expression of *GGPPS* at 20 DAS and 40 DAS could explain the increased chlorophyll a and chlorophyll b changes in the inner leaves, although regulation of chlorophyll synthesis in the outer leaves at other levels of regulation such as post-transcriptional regulation should also be considered [24].

The phytoene synthase enzyme that converts gerangerandyl diphosphotase into phytoene (encoded by the two *PSY* genes in lettuce) is the first committed key-limiting step in carotenoid biosynthesis [2] and controls the expression of genes involved in carotenoids biosynthesis [41]. *PSY2* is highly expresses in the inner leaves at the seedlings stage of lettuce plants, while *PSY1* is highly expressed at 40 DAS and Comm stage in the inner and outer leaves, respectively.

In the subsequent steps of phytoene into lycopene transformation, plants employ two desaturases, phytoene desaturase (PDS), forming 11 and 11’ cis double bonds, and ζ-carotene desaturase (ZDS), forming 7 and 7’ cis double bonds [42]. The upregulation of any of the desaturation-related genes impacts the homeostasis between them, and the expression level is upregulated in adjustment [2,24]. Similarly, we observe that in lettuce, *PDS* and *ZDS* were highly upregulated in the inner leaves at the seedling stage and 40 DAS, while outer leaves had lower relative expression level. In the next stage the carotene isomerase (*CRTISO*), upregulated at the seedling stage, is employed to complete conversion into all-trans-lycopene containing 11 C-C bonds essential for light energy absorption and carotenoids functioning [43].

Cyclization of all-trans-lycopene bifurcates the pathway into two branches: the β-β branch leading to b-carotene, violaxanthin and, finally, abscisic acid; and the ε-β branch leading to lutein biosynthesis [25]. In lettuce and some other plant species, the cyclization could have the third ε-ε-branch, leading to lactucaxanthin biosynthesis (Appendix A) [27,28]. The ε-ε-branch was out of the scope of this research, as we mainly focused on β-carotene and lutein biosynthesis and transcriptional regulation.

While key carotenoid biosynthesis genes (*DXS, DXR, CRTISO*) as well as first enzymes of ε−β branch were highly upregulated at 20 DAS, the lutein content was the lowest at this developmental stage across all genotypes, ranging from 13 to 21 μg g^−1^ FW (Figure 2). The only enzyme of the ε−β branch wherein upregulation was corresponding a 2–2.1-fold lutein increase at 40 DAS was *CHY-ε*, the last enzymatic stage of lutein biosynthesis, wherein relative expression at 40 DAS was increased 1.95 ± 0.48-fold as compared to the 20 DAS relative expression level (Figure 6).

In the β-β branch, the enzyme lycopene β-cyclase (LCYB), encoded in lettuce by *LCY-β1* and *LCY-β2*, catalyzes the formation of two β-rings at both ends of the lycopene molecule, resulting in β-carotene production. Again, the highest expression of *LCY-β1* and *LCY-β2* was observed at 20 DAS (seedling stage), while β-carotene accumulation was on average 3-fold increased (Figure 6) at 40 DAS (up to 288.6 ± 19.7 μg g^−1^ FW, Figure 2A) compared to at 20 DAS (42.5 ± 6.4 μg g^−1^ FW to 65.2 ± 9.6 μg g^−1^ FW, depending on genotype, Figure 2A). At the Comm stage, it had fallen to just half its 40 DAS levels, a trend observed in both inner and outer leaves.

Next, we performed a co-expression correlation analysis using *LCY-β1* and *LCY-β2* genes as drivers to determine the level of co-expression that genes encoding lycopene β-cyclase share with all other analyzed genes (Appendix A). The highest significant co-expression pattern was observed for the β-ring hydroxylase (*CHY-β*) gene (*p* < 0.01, Appendix A), encoding the enzyme involved in β-carotene conversion to zeaxanthine, and for zeaxanthin epoxidase (*ZEP*)/violaxanthin deepoxidase (*VDE*) genes, regulating zeaxanthine biosynthesis [2,25,44].

The second pathway after bifurcation, the ε-β branch is controlled by LCYE (Lycopene Epsilon Cyclase, encoded by single gene *LCY-ε*) and Lycopene Beta Cyclase [25]. The final step of ε- and β-ring hydroxylation of α-carotene are catalyzed by CHYE, ε-ring hydroxylase, yielding lutein. The expression correlation analysis using ε-ring hydroxylase (*CHY-ε*) as the driver gene revealed a significant co-expression pattern between *CHY-ε* and the *LCY-ε* and *LCY-β2* genes (*p* < 001). *CHY-ε* was up-regulated in the inner leaves at all developmental stages and in the outer leaves at the commercial stage, which corresponded to the increased of lutein accumulation at 40 DAS and 60 DAS.

We noted that *CRTISO*, *CHY-β* and *VDE/ZEP* were co-expressed if *CHY-ε* was used as the driver (Appendix A), which could indicate that both pathways were controlled by levels of their common precursor (all-trans-lycopene), and upregulation of one branch would lead to reduced biosynthesis of compounds in the second branch.

In summary, the co-expression analysis suggested that to ensure high carotenoids content in leaves, the metabolites sink from β-carotene and lutein needs to be regulated. For example, by preventing zeaxanthin accumulation (e.g., via *CHY-β* gene silencing or overexpression of *VDE*), we could prolong accumulation of β-carotene and lutein in the leaves and, as a result, to delay the senescence symptoms appearance in lettuce leaves.

## 3. Discussion

### 3.1. Changes in Carotenoid Accumulation Related to Lettuce Genotype and Developmental Stages

The regulation of carotenoid accumulation occurs at multiple levels: transcriptional, post-transcriptional, post-translational and storage/degradation (see review in [19]). Here, we focused only on the transcriptional regulation of carotenoid biosynthesis in six lettuce cultivars during three developmental stages, since if transcriptional regulation on carotenoids biosynthesis is only part of the whole picture, it should not be neglected. In leafy vegetables, the biosynthesis and accumulation of carotenoids occurs during plant cultivation. Carotenoids are accessory pigments, are involved in light harvesting and are essential for effective photosynthesis. The steady-state concentrations of carotenoids in leaves are considered to result from a balance of their biosynthesis and degradation [45], and changes in carotenoid content are usually associated with photosystem imbalance [46]. Carotenoid content can change in response to the type and source of light and external factors like temperature [2]. Green lettuce cultivated in greenhouses supplemented with UV-A and UV-A+UV-B has higher β-carotene and lutein concentrations compared to those cultivated without UV- radiations [47,48], as found in our experiments. In this study, metabolite data obtained by reverse HPLC was combined with gene expression data generated by qRT-PCR analysis to characterize carotenoids biosynthesis during lettuce plant development across cultivars which display contrasting green color intensity.

To better understand the relationship between expression levels of carotenoid biosynthesis pathway genes and carotenoid levels, we investigated the expression levels of gene-encoding enzymes involved in the control of carotenoid accumulation in inner and outer leaves of the lettuce plants during different developmental stages, after identifying these from the lettuce genome. Lutein and β-carotene levels were measured in different-age leaves for all tested cultivars. The levels of these carotenoids at commercial size were comparable with carotenoid levels detected in other green leafy vegetables, such as basil, kale and Brassica oleracea cultivars [21,22,49]. In a previous study [23], levels of β-carotene and lutein were quantified in three of the lettuce genotypes used in our study. We report similar levels to those detected by Mou [23], except for the cultivar Salinas 88, which we found to contain almost ten times greater amounts of both carotenoids in leaves at similar growth stages, which could be attributed to our soil mineral composition, light intensity and the use of outer leaves which had higher green color intensity than the leaves used previously [23].

The lettuce cultivars investigated displayed a significant variation in chlorophyll, β-carotene and lutein levels. It was previously reported that green color intensity could serve as a proxy indicator of β-carotene concentration in lettuce [23]. The chlorophyll a is synthesized through geranylgeranyl diphosphate (GGPP) (Appendix A), and mutations in GGPPs lead to alteration of both chlorophyll and carotenoids content in plants [19]. The Pearson correlation analysis (Appendix A) indicated that the *PSY1* gene in inner leaves at the commercial stage is positively correlated with total chlorophyll in outer leaves (r = 0.85, *p* < 0.05), which suggests an indirect relation to β-carotene. In our study, we also observed negative correlation between chlorophyll levels and β-carotene and lutein contents (Figure 3B). The lutein concentrations had weak negative correlation with total chlorophyll content in leaves (R^2^ = −0.53, *p* > 0.05, Figure 3A), meaning that lutein content could not be predicted by color intensity levels, appealing though this might be for consumers. For example, while ‘Dragoon’ and ‘Salinas 88’ had different leaf color between inner and outer leaves, these cultivars had the same concentration of lutein in the inner and outer leaves, indicating that that lutein concentrations are unrelated to leaf color intensity.

During senescence, the chlorophylls are degraded (Figure 2), and chlorophyll a is decomposed into phytol and pheophorbide a [19]. In most lettuce cultivars, we found carotenoid and lutein concentrations to peak at 40 DAS (Figure 2), while at the usual commercial stage of harvesting, 60 DAS, when chlorophyll was already degraded, carotenoid levels had also started declining, leading to accumulation of abscisic acid (Figure 6). A similar pattern has been observed previously in *Arabidopsis thaliana* leaves, where a 50% increase in carotenoid levels was observed in newly-formed adult leaves compared to older ones [50], and therefore, carotenoid accumulation was associated with the developmental stage.

Leaf maturation is a multicomponent process that includes morphological and cellular changes, like the increase in chloroplast size and chloroplast density per cells, and is tightly regulated at the molecular level. For all lettuce cultivars tested, chlorophyll and carotenoid concentrations varied according to plant development stage. Thus, chlorophyll and chlorophyll b were the highest at the seedling and 40 DAS stages, followed by reduction in all cultivars (Figure 2) when plants reached the commercial size. Although all photosynthetic pigments will eventually degrade, chlorophyll was lost more rapidly than β-carotene, while lutein remained relatively stable (Figure 2) like in senescent Arabidopsis and tobacco leaves [48,50,51]. The role of ABA in carotenoids degradation is supported by co-expression of *CHY-β* gene, encoding the enzyme involved in β-carotene conversion to zeaxanthin with zeaxanthin epoxidase (ZEP) gene, essential for ABA biosynthesis (Appendix A). The transition of chloroplasts into chromoplasts results in an enhanced storage capacity for β-carotene [52] and accumulation of carotenoids. In all lettuce cultivars, chlorophyll concentrations were lower in the inner leaves, and the peak in lutein and β-carotene accumulation at 40 DAS vs 20 DAS could be consequence of this transition. In the outer leaves, this reduced chlorophyll content was noticeable only in Grand Rapid and Veronica, which both have similar plant architecture. In both cases, reduction in chlorophyll a and chlorophyll b was associated with lutein and β-carotene accumulation (Figure 2). The reduction of β-carotene content at 60 DAS as compared to 40 DAS, and similar or higher levels of lutein at 40 and 60 DAS in the inner and outer leaves could be explained by the cleavage of β-carotene branch carotenoids to produce strigolactones and ABA, which further promote leaf senescence [24,53].

### 3.2. Relationship between Biosynthetic Gene Expression and Carotenoid Concentration

Upon exposure to light, both carotenoids and chlorophyll are synthesized, and co-expression of chlorophyll- and carotenoid-related genes is evident in many plants [24,40]. The light-green leaf color in the Verônica and Grand Rapids cultivars was consistent with the lower amounts of carotenoids detected, while the dark green color observed in ‘Dark Land’ and ‘Parris Island’ was consistent with higher concentrations of detected carotenoids. The relationship between color intensity and carotenoid levels was also evident for ‘Dragoon’ and ‘Salinas 88’, which displayed significant differences in carotenoid concentrations in outer and inner leaves. This correlation between carotenoid levels and green color intensity is likely related to the protective role of carotenoids in the photosynthetic matrix, since carotenoids are essential to protect chlorophyll from degradation [48,54,55].

The rate-limiting enzyme, DXS, is essential for controlling carotenoid biosynthesis flux [29]. The negative correlation between DXS levels and total chlorophyll is consistent with reduced levels of expression at later stages for a gene which is involved in the first step of the biosynthetic pathway (Figure 6, Appendix A). Indeed, DXS expression levels were significantly reduced in all genotypes compared to other tested genes, suggesting a significant reduction in pigment synthesis in older tissues (Figure 6). Similar DXS reduction during developmental stages was observed in red pepper fruits [56].

The expression of *PSY1* and *PSY2*, which encode the PSY enzymes that catalyze the first committed step in carotenoid biosynthesis, was also found to be positively correlated with chlorophyll and carotenoid accumulation (Appendix A). A similar relationship between carotenoid synthesis and *PSY1* expression has been shown in *Brassica napus* and in *Arabidopsis* [57,58] and *PSY1* expression in ripe tomato fruits [59].

A correlation of *CRTISO* and *LCY-β2* transcript levels with chlorophyll and carotenoids was not observed during the early stages of development, but it was detected at the commercial size stage (Figure 6, Appendix A). A similar pattern was observed in apple fruits, where the expression of some genes predominates after pollination but is reduced before the ripening stage [11]. The differential expression of *CRTISO* between inner and outer leaves at the commercial stage suggests that this gene could be associated with a ‘pause’ in carotenoid synthesis. In inner leaves, the *CRTISO* gene showed higher levels of expression compared to outer leaves while the carotenoid synthesis was found to be constant, following the same pattern of accumulation between cultivars (Figure 2, Appendix A). In contrast, in outer commercial leaves, the *CRTISO* expression was different, with higher transcript levels detected in dark green cultivars. This observation was confirmed by the positive and negative correlations between the *CRTISO* gene expression levels and carotenoids accumulation in inner and outer leaves, which could help to explain the lower carotenoid levels detected in the plants at the commercial stage (Figure 2, Appendix A).

The difference in carotenoid accumulation between lettuce cultivars could also potentially be explained by degradative enzyme action. Some early pathway genes displayed a reduction in total transcript levels at the commercial size stage (Figure 4, Appendix A). An increase in total transcript levels of VDE observed at 60 DAS (when cultivars had reduced carotenoid concentration compared to 40 DAS) could suggest that carotenoid degradation had occurred in the internal leaves, resulting in total carotenoid concentration being reduced at 60 DAS. Our results also indicate a reduction in β-carotene levels in the inner leaves of ‘Salinas 88’ and ‘Dragoon’ when they reached the commercial size. The production of carotenoids after synthesis of phytoene in these cultivars may be affected by cleavage enzymes during carotenoid synthesis. A similar mechanism has been suggested for chrysanthemum, in which white cultivars had similar expression levels of gene-encoding intermediate enzymes of the pathway as in a yellow cultivar but increased expression of carotenoid cleavage genes [16].

Total carotenoid levels in plant tissues result from an interplay between carotenoid biosynthesis, sequestration, storage and degradation. Key genes which could affect carotenoid synthesis have been identified. Transcript levels of *LCY-ε* can be directly related to increased lutein concentrations in several species [11,15,16]. In our experiments, the pattern of transcript level changes was similar among all cultivars, where critical genes had similar levels of expression. Hence, to explain the different levels of carotenoids and chlorophyll in lettuce cultivars, additional mechanisms of regulation beyond transcript levels should be considered. 

In expression correlation analysis using *CHY-ε* as the driver gene for lutein biosynthesis, we found significant correlation between *CHY-ε* and *LCY-ε* expression, leading to lutein accumulation (Appendix A). We also found co-expression between *LCY-ε* and *DXS* genes which regulate the carotenoids biosynthetic pathway. This could suggest that the synthesis of lutein is predominantly controlled by the expression of early pathway genes and that a later degradation process occurs during plant development. The *LCY-β1* and *LCY-β2* genes, encoding the LCYB enzyme which participates in controlling the flux of metabolites through the ε−β and β−β branches [58,60,61], were also highly expressed at early stages of development. The lack of correlation between LCYB expression and β-carotene accumulation points towards the existence of an upper threshold in *LCY-β1* and *LCY-β2* transcript levels and the accumulation of β-carotene [25]. The downregulation of LCY-β2, and reduced expression of *LCY-β1* at both 40 and 60 DAS in the inner leaves as compared to the seedling stage, opposite to lutein and β-carotene accumulation, suggests different levels of carotenoid regulation at the later stages of plant development—for example, carotenoid biosynthesis via the retrograde signaling pathway from plastids to the nucleus [52]. The co-expression with *CHY-β* gene suggests that flux toward ABA biosynthesis is an important regulatory mechanism that controls β-carotene biosynthesis and degradation in adult leaves.

To summarize, in our research, we compared gene expression patterns to carotenoids (β-carotene and lutein) and chlorophyll metabolic changes in the inner and outer leaves throughout the plant life cycle to understand if carotenoids accumulation is differently regulated in plants with distinct architecture and leaf color. We did not identify master regulators responsible for divergent expression of carotenoid biosynthetic genes leading to variety led changes in carotenoid contents. However, our data prompt to a clear natural variation in carotenoid biosynthesis pathway, likely controlled at least partly at the gene expression level. A major finding in our paper is the identification of three genes (*GGPPS*, *PSY1* and *LCY-β1*) whose transcript levels can be used to predict carotenoid levels at the commercial harvested stage for lettuce. Our results also suggested that plants selected based on the delayed leaf senescence phenotype will also have a higher concentration of β-carotene and lutein for extended period.

## 4. Materials and Methods

### 4.1. Plant Material and Growing Conditions

Six commercial lettuce cultivars were chosen for analysis based on differences in plant architecture and color intensity: Dark Land Cos MT (romaine type, dark green), Dragoon (mini romaine, dark green), Grand Rapids (batavia type, light green), Parris Island (romaine type, medium green), Salinas 88 (crisphead, medium-green intensity) and Verônica (batavia, light green) (Appendix A). Plants were grown at the University of Galway, Ireland, in a plant growth chamber under fluorescent lighting with light intensities of 300 μmol·m^−2^·s^−1^ under long-day conditions (16 h light–8 h dark). Lettuce seeds were sown on soil in 200 mL pots and transplanted to three-liter pots at the seedling size (20 days after sowing (DAS)). The experiments were conducted in a complete randomized design with three replications and three plants per genotype per replication. All plants were grown at the same time and in the same chamber, with the same light intensity and temperature to minimize environmental variability. Leaf samples of each cultivar were harvested at three different stages, corresponding to seedling size (transplanting time), 40 DAS and at commercial size (60 DAS). Samples at each plant stage represent an independent group of plants. At the 40 and 60 DAS stages, the outer leaves (first external leaves) and inner leaves (a leaf from the seventh inner layer of leaves from the outside of the head) were sampled. In each lettuce plant, sampling was done on three different leaves around the plant in the same location in each outer or inner leaf, corresponding to a total of three pooled samples per plant, and nine pooled samples per sample. The harvested samples from each sampling time and leaf stage were frozen immediately and stored at −80 °C until analysis.

### 4.2. Carotenoid Extraction and HPLC Analyses 

Carotenoid pigments were extracted following procedures described by Norris [62] with some modifications. Leaf tissue (0.3 g) was ground under liquid nitrogen in a porcelain mortar and transferred to a 2 mL centrifuge tube with one glass bead. Then, 200 µL of 80% *v*/*v* acetone was added before adding ethyl acetate (200 µL), and the tubes were agitated at 30,000 rpm for 1 min in a tissue-lyser (RETSCH MM200—Qiagen, Manchester, UK). Water (140 µL) was added, and the mixture was agitated again at 30,000 rpm for 1 min and then centrifuged at 15,800× *g* for 5 min in a microcentrifuge (Heraeus Fresco 17—ThermoScientific, Ireland). The upper phase, containing carotenoids, was then transferred into a new tube. The samples were extracted at least three more times, adding 200 µL of ethyl acetate, agitating, centrifuging at 15,800× *g* for 5 min and removing the upper phase until the precipitate did not have any visible green color. The combined ethyl acetate phases were vacuum dried in a centrifugal evaporator (miVac—GeneVac SP Scientific, UK). The dried samples were subsequently redissolved in 1.5 mL of 0.8% of BHT/acetone [17] and analyzed by the reverse phase of high performance liquid chromatography (HPLC). The HPLC system (Alliance, Waters Co., Milford, Mass.) consisted of a separation unit (model 2695), YMC 4.6 × 10 mm C30 guard cartridge and YMC RP C30 column (3 μm, 250 × 4.6 mm—YMC, Wilmington, North Carolina, USA). The column temperature was 25 °C; samples were kept in a 4 °C sample cooler, and a 50 µL aliquot was injected into a 1 mL min^−1^ flow rate. The elution was performed using a mobile phase comprising solvent A (MeOH), solvent B [H_2_O/MeOH, 20:80) containing 0.2% *w*/*v* ammonium acetate] and solvent C (tert-butyl methyl ether). The elution gradient was a reduced-time version of that described by [11]. The gradient started with 95% A/5% B for 2 min, decreasing to 80% A/5% B/15% C between 2 and 10 min, decreasing to 30% A/5% B/65% C by 15 min, decreasing to 25% A/5% B/70% C at 20 min, and returning to 95% A/5% B at 25 min. The β-carotene and lutein picks were identified by comparing the retention time (RT) and absorption spectra of individual peaks within each sample with the β-carotene and lutein standards RT (Sigma-Aldrich, Arklow, Ireland). To ensure that RT was not changed/masked by other plants metabolites, β-carotene and lutein standards were also spiked into plant samples followed by RT comparison to standards containing no plant extracts (Appendix A). Plant samples extraction losses were determined from the initial concentration of the internal standard trans-β-Apo-Carotenal (Sigma-Aldrich, Ireland). The linear portions of the standard curves were used to convert the integrated areas of both β-carotene and lutein as µg per g of fresh weight.

### 4.3. Chlorophyll Extraction and Quantification 

Chlorophyll pigments were extracted by crushing the samples in a tissue-lyser (RETSCH MM200—Qiagen, UK) in tubes with one glass bead for 1 min at 30,000 rpm. Then, 2 mL of 96% *v*/*v* ethanol was added, and the samples were agitated again for 1 min at 30,000 rpm and left for 24 h at 4 °C in a 2 mL centrifuge tube. Next, the absorbance of the samples was measured using spectrophotometer (Nanophotometer—Inplen, Germany) at 649 and 665 nm wavelengths. The chlorophyll a (*Chla*) and *b* (*Chlb*) concentrations were calculated by the following formulas:(1)$${Chl}_{a}={13.95A}_{665}-{6.88A}_{649}$$
(2)$${Chl}_{b}={24.96A}_{649}-{7.32A}_{665}$$

The total chlorophyll was calculated by the sum of *Chla* and *Chlb*, and the values were determined as µg g^−1^ of fresh leaves. 

### 4.4. RNA Extraction and cDNA Synthesis

Total RNA was extracted from leaf samples using the Isolate II RNA Mini Kit (Bioline, UK). Samples were crushed with glass beads in a tissue-lyser (RETSCH MM200—Qiagen, UK) for 1 min at 30,000 rpm. Lysis buffer was added to the ground tissue followed by vigorous vortexing. The lysate was loaded in the filter tube column and centrifuged for 1 min at 11,000× *g* (Heraeus Fresco 17—Thermo Scientific, Dublin, Ireland) to isolate impure particles. In the filtered samples, 70% *v*/*v* ethanol was added, followed by vortexing and loading in a second filter tube column. The samples were centrifuged for 30 s at 11,000× *g* and washed two times with wash buffer and dried. Total RNA was eluted with RNase-free water, followed by DNase treatment according to the manual (AMPD1, Sigma-Aldrich, Ireland). cDNA was synthesized from total RNA (0.5–1 μg) using a SensiFAST cDNA Synthesis kit (Bioline, UK) according to the manufacturer’s protocol. The reaction components were 5× TransAmp Buffer and Reverse Transcriptase. The reaction was incubated at 25 °C for 10 min (primer annealing), 42 °C for 15 min (reverse transcription) and 85 °C for 5 min (inactivation). 

### 4.5. Quantitative Real-Time PCR Analysis

Lettuce genes involved in the carotenoid biosynthetic pathway were identified through searches of the Expressed Sequence Tag (EST) database deposited in NCBI (National Center for Biotechnology Information) with tBlastn where known carotenoid biosynthetic proteins from chrysanthemum (*Chrysanthemum indicum L.*, Asteraceae), apple (*Malus domestica Borkh.*, Rosaceae) and Arabidopsis (*Arabidopisis thaliana (L.)* Heynh., Brassicaceae) were used as queries. Primers were designed for the 17 target carotenoid biosynthesis genes and for a lettuce UBIQUTIN housekeeping gene (LsUBI; Appendix A) using Quantprime (http://www.quantprime.de/main.php?page=home, last accessed on 17 June 2023). cDNA was generated from the same samples that had been used for carotenoid and chlorophyll quantification, and expression levels of the 17 carotenoid biosynthesis genes were determined by Quantitative real-time PCR (qRT-PCR) using a CFX96 Real-Time system (BioRad, Watford, UK). The SYBR Green master mix was used following manufacturer’s protocol with minimum modifications. cDNA templates were diluted 1:4 times and used in a 5 µL final volume reaction (1 µL of cDNA template, 0.25 µL of each primer, 1 µL of water and 1 µL of SYBR green). For each sample, three technical replicates were prepared with three negative controls per plate. The PCR reaction conditions were 95 °C for 10 min (preincubation), followed by 40 amplification cycles (95 °C for 15 s, 60 °C for 30 s and 72 °C for 30 s). A melting curve analysis with continuous fluorescence measurement during the 65–95 °C melt was generated after the amplification reactions. Data were analyzed using BioRad CFX manager software (BioRad, UK). The expression level of each gene was normalized to the lettuce UBIQUTIN gene. Standard deviation was calculated using the three biological repetitions for each sample.

### 4.6. Bioinformatic Identification of Candidate Genes

Candidate genes were identified by tBLASTn using reference sequences from chrysanthemum (which is in the same family, Asteraceae, as lettuce), apple and A. thaliana as queries against lettuce EST databases (NCBI) and genome project sequence (https://lgr.genomecenter.ucdavis.edu/Links.php, last accessed on 17 June 2023).

### 4.7. Statistical Analysis

The transcriptome and metabolites levels (carotenoids and chlorophylls) were compared by one-way ANOVA followed by Tukey HSD post hoc test assuming normal distribution and variance homogeneity. If the assumption did not apply, a Kruskal–Wallis one-way ANOVA on ranks was performed. Significant differences were considered at *p* ≤ 0.05 and were indicated by different letters or asterisks. Outlier identification was performed in RStudio (2023.03.0+386 “Cherry Blossom” Release for MacOS) using Grubbs test (package “outliers”). Metabolite levels are presented as mean concentrations ± standard error unless otherwise stated. Fold changes of carotenoids and chlorophyll change are normalized to the corresponding levels in the Grand Rapid variety. Linear regression analysis was performed in SPSS 28.0 to predict the effect of chlorophyll and lutein on β-carotene content in leaves. Pearson correlation analysis to identify correlations between chlorophyll and carotenoids levels was performed using the heatmaply package in R. Principle Component Analysis (PCA) was conducted in RStudio (package ggfortify) by using ESTs or normalized carotenoids levels as variables. The heatmap analysis was performed with z-normalized data using pheatmap package in RStudio. The variables were clustered according to the inner or outer leaves samples, time of sampling and genotype. The heatmap data is presented as row normalized. A Pearson correlation analysis between gene expression level was performed in SAS Statistical Analysis Software STAT 14.1 (www.sas.com, last accessed on 17 June 2023) and tested for statistical significance using “proc corr”.

## 5. Conclusions

To date, carotenoid biosynthesis and degradation pathways have been mostly elucidated using *Arabidopsis thaliana* and other models. In addition to specific carotenoid degradation enzyme activities, e.g., Carotenoid Cleavage Dioxygenases, carotenoids can be degraded non-enzymatically due to high levels of lipid peroxidation and oxidative stress in senescing tissues [24,63]. Simultaneous detection of gene expression levels and carotenoid levels across contrasting color (and architecture) lettuce genotypes has allowed us to reveal the relationship between gene expression and carotenoid accumulation in inner and outer leaves at different stages of the lettuce life cycle. Our study demonstrates that carotenoids and chlorophyll accumulation levels vary during development and leaf age of lettuce plants. Our results also suggest that expression levels of genes encoding key enzymes of the carotenoid biosynthetic pathway may cause these differences and that strategies to manipulate carotenoids genes and carotenoid content in plants must be evaluated case by case. Our study provides a basis for further determination of the control of carotenoid biosynthesis in a crop species so that nutritionally improved lettuce varieties can be developed. 

## Figures and Tables

**Figure 1 ijms-24-10310-f001:**
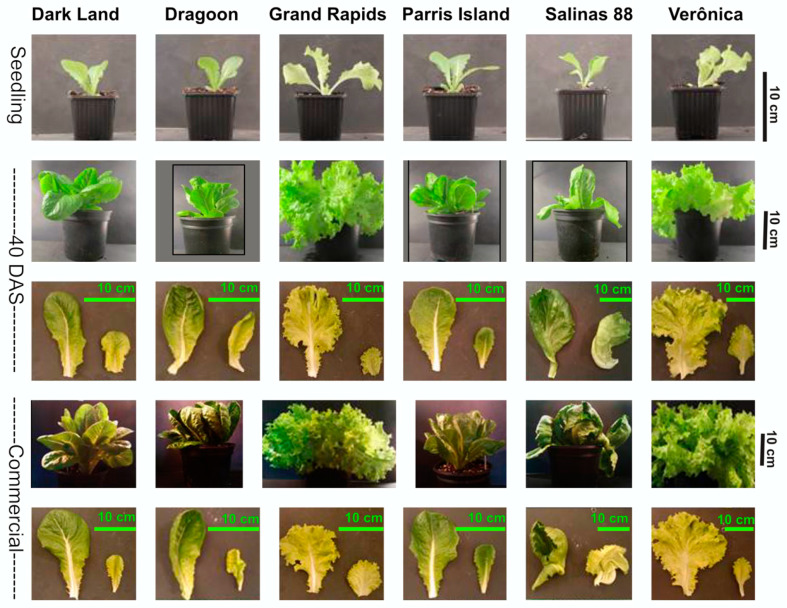
Development of six lettuce cultivars growing under the same environmental conditions at seedling stage: 20 days after sowing (DAS), 40 DAS and at commercial size (60 DAS).

**Figure 2 ijms-24-10310-f002:**
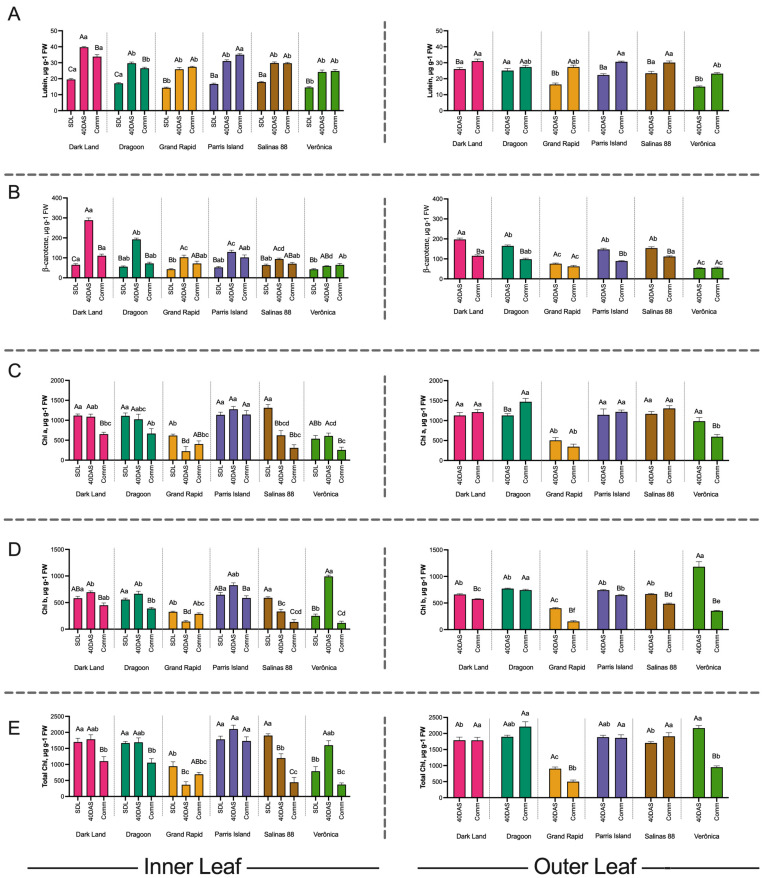
Analysis of chlorophyll and carotenoids in the inner and outer leaves of six lettuce cultivars. (**A**) lutein, (**B**) β-carotene, (**C**) Chla, (**D**) Chlb and (**E**) total chlorophyll content in the inner and outer leaves of six lettuce cultivars at seedling stage (SDL, 20 Days after sowing (Das)), 40 DAS and at commercial size (60 DAS). Error bars are standard deviation of the mean from three biological replicates, each being the pooled sample of at least three technical replicates, at *p* = 0.05 level. Bars with similar capital letters for each cultivar in each leaf position at different time points are not significantly different (*p* ≥ 0.05), using one-way ANOVA analysis followed by Tukey pos hoc test. Bars with similar small letters at the same time point in each leaf position between cultivars are not significantly different (*p* ≥ 0.05), using one-way ANOVA analysis followed by Tukey pos hoc test.

**Figure 3 ijms-24-10310-f003:**
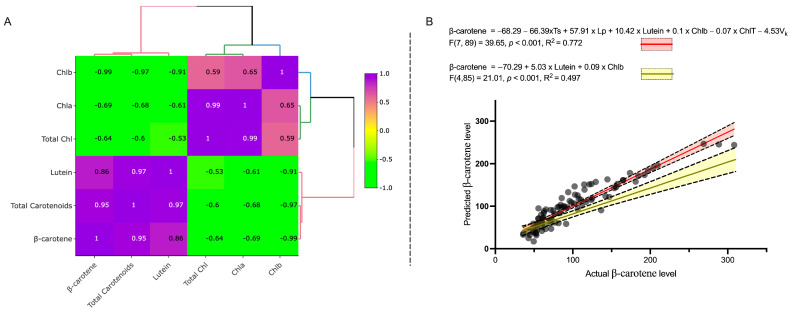
Pearson correlation on normalized metabolites in six lettuce varieties from the inner and outer leaves during three developmental stages revealed correlation between β-carotene and lutein and confirmed inverse correlations between carotenoids and chlorophyll contents in lettuce plants (**A**). Regression analysis showing that β-carotene content can be predicted by lutein content in plants (blue line) and strongly depend on plant genotype (Vk), plant developmental stages (Ts), sample location (Lp) and chlorophyll b level (**B**).

**Figure 4 ijms-24-10310-f004:**
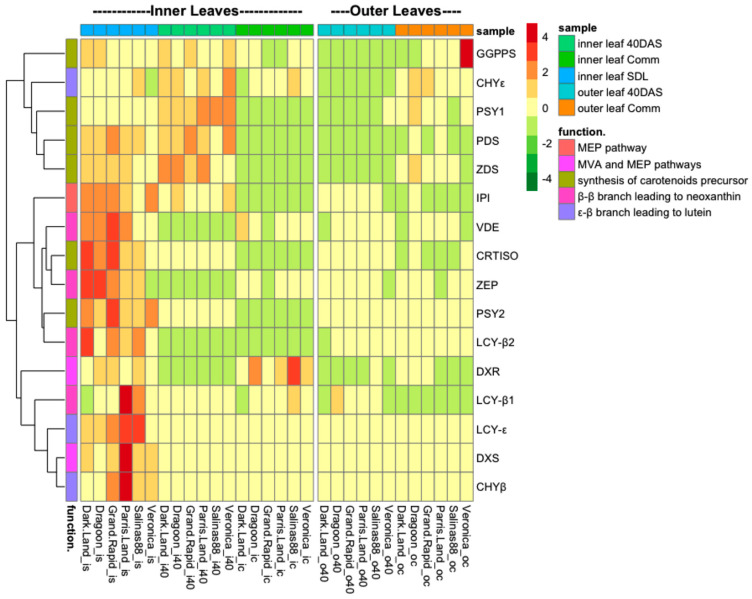
Analysis of carotenoids biosynthesis genes expression in the inner and outer leaves of six lettuce cultivars was measured at seedling stage (20 days after sowing (DAS)), 40 DAS, and commercial size (60 DAS). Gene expression was measured relative to *UBIQUITIN* (*n* = 3 biological replicates for each sample).

**Figure 5 ijms-24-10310-f005:**
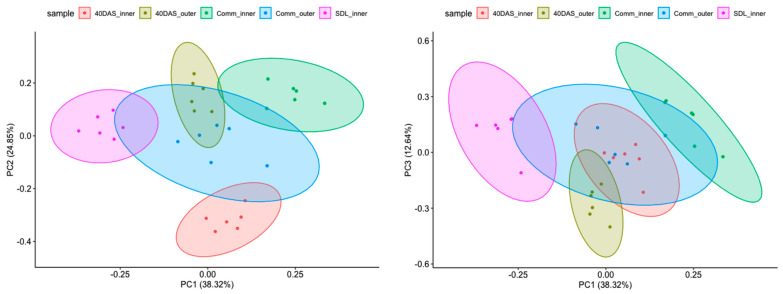
PCA analysis (PC1:PC2 and PC1:PC3) of gene expression at different developmental stages in inner and outer leaves of six lettuce cultivars.

**Figure 6 ijms-24-10310-f006:**
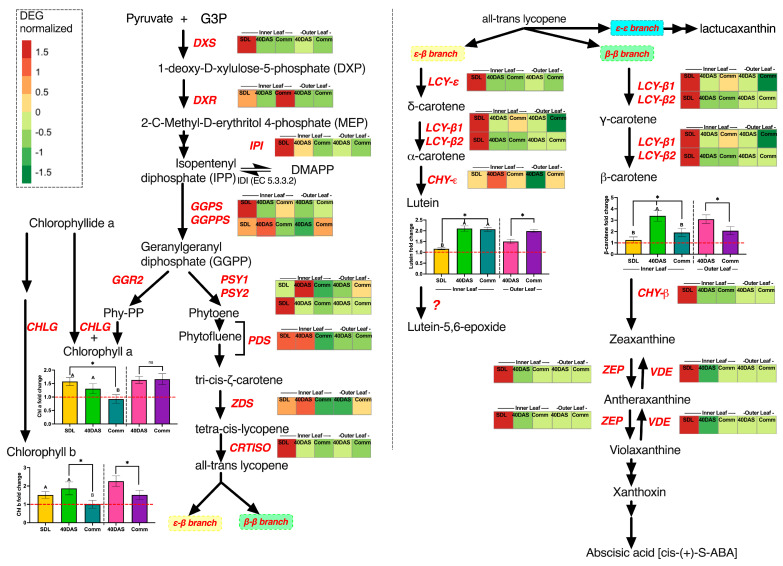
Metabolomic and transcriptomic pathways of carotenoid biosynthesis in *Lactuca sativa*, Asteraceae. Genes that encode pathway enzymes are shown in red italic letters. The heatmap is the normalized average of all six varieties at three developmental stages in inner and outer leaves. Abbreviations: GA3P, glyceraldehyde-3-phosphate; DXS, 1-deoxyxylulose 5-phosphate synthase; DOXP, D-1-deoxyxylulose-5-phosphate; DXR, 1-deoxyxylulose 5-phosphate reductoisomerase; MEP, 2-C-methyl-D-erythritol-2,4- cyclodiphosphate; IPP, isopentenyl diphosphate; IPI, IPP isomerase; GGPP, geranylgeranyl diphosphate; GGPS, GGPP synthase; PSY1, phytoene synthase; PDS, phytoene desaturase; ZDS, ζ-carotene desaturase; CRTISO, carotenoid isomerase; LCY-B, lycopene β-cyclase; LCY-E, lycopene ε-cyclase; CHYB, β-ring hydroxylase; CHYE, ε-ring hydroxylase; ZEP, zeaxanthin epoxidase; VDE, violaxanthin deepoxidase; CHLG, Chlorophyll synthase; GGR2, geranylgeranyl reductase2. Bars with similar capital letters at different time point means that samples are not significantly different (*p* ≥ 0.05), using one-way ANOVA analysis followed by Tukey pos hoc test. * Means that samples are significantly different based on the *t*-test.

## Data Availability

The data available upon request.

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
