# Peer review of "Regulation of Carotenoid Biosynthesis and Degradation in Lettuce (Lactuca sativa L.) from Seedlings to Harvest"

_ijms, 2023, doi:10.3390/ijms241210310_

Round 1

Reviewer 1 Report

Review report of the MS_ijms-2399257-peer-review-v1

The manuscript ‘Regulation of carotenoid biosynthesis and degradation in lettuce
(Lactuca sativa L.) from seedlings to harvest:
’ Authors by Brychkova et al. is well-written. However, I have a few suggestions which could improve the manuscript.

Abstract: Please re-arrange the sentence to improve the flow of the paragraph.

Line 24 is not required. Please start with the line 25. Please write the sentence directly…’Lettuce (Lactuca sativa L.) is one of the commercially important leafy vegetables worldwide’.

Line 26: At the time of harvest

Line 29: Performed on the inner

Line 31: affecting the accumulation

Line 32/585: @-carotene; Is it β-carotene? Please use the correct symbol.

Please place line 31-33 “our results….commercial cultivars” after the line ‘Mathematical analysis…..cultivar’.

Line 33: Please revise ‘Mathematical analysis’ as ‘Statistical analysis’

Please cross-check the English language, grammar, and punctuation throughout the text.

Line 35-39: Please re-write the sentence with the key findings of the work with their implications.

Line 55: …a better understanding.

Line 60-65: Please re-write clearly with small sentences for better understanding.

Line 89: ζ-carotene, please use the correct symbol.

Line 99: Please introduce the full form of DXS and DXR; ZISO, CRTISO, and LCY-ε on their first appearance.

Line 129: Either use ‘Moreover’ or ‘Despite’

Line 132-138: Please write the main objective of the study; Please mention our aim/objective to investigate/relate instead of we investigate/relate.

Line 175-178: ‘nearly tripled’..please revise as ‘three fold’. In the unit, please revise ‘Fw’ as ‘FW’ throughout the text. ‘Halved’ may be revised as ‘reduced by 50%’

Figure 2 is well-presented. It is better to split the Figure into two Figure 2a-e: (a) β-carotene, (b) lutein, (c) Chla, (d) Chlb and (e) total chlorophyll content; in outer and inner leaves; and Figure 3a-b: correlation and regression.

Please make the result and discussion section ‘justified text’

Section 2.2: Please quantify the result. Pl mention the relative expression (-fold increase/decrease). Please avoid writing the materials/methods.

Line 239-240: Declined progressively….pl quantify.

Figure 3 may be split into two; Heat map, and PCA.

Figure 4: Tukey’s significance level ‘Abc’ is missing.

The results contain more discussion. Please mention the quantifiable results in the result section, and the rest can be discussed in the discussion section.

Please conclude the investigation with the key findings (quantifiable) and the future implications.

Line 469-471: The genotype description may be presented in a table with the location.

Line 476: ‘three plants per plot’ may be written as ‘three plants per genotype per replication’

Line 478: Pl mention the light intensity

Line 486: Why the authors mentioned ‘Plot’ for an pot experiment?

Line 493: Please don’t start the sentence with a number 140…please revise as Water (140 µL) was added.

Line 498: Please revise15.800gn as 15.8g

Line 507: Please revise ‘1 mL min-1’ as ‘1 mL min-1’..Please revise throughout the text.

Line 509: Pl revise [H2O/MeOH, 20:80] as [H2O/MeOH, 20:80]

Line 510-514: Please re-write

Line 527-528: Please number the equations as …[1]…[2]

Line 588: Please mention the version of Rstudio (package ggfortify) [if any].

References may be arranged as per the journal pattern. Please cross-check the references cited with the list and text.

Although the overall grammar score is satisfactory, the authors are requested to polish the language, grammar, and punctuation again while doing the minor revision.

The manuscript may be accepted with minor corrections.

Good luck with the revision.

Although the overall grammar score is satisfactory, the authors are requested to polish the language, grammar, and punctuation again while doing the minor revision.

Author Response

Thank you very much for reviewing our manuscript. Please see the attachment for point-by-point response to your comments.

Reviewer 2 Report

The publication presents chlorophyll and some carotenoid contents in parallel with some transcript levels of genes for enzymes of the plastid-localized IPP-synthesizing and carotenoid biosynthesis pathway in outer and inner leaves of various lettuce cultivars (lactuca sativa L.). In a purely descriptive work, HPLC data of carotenoids were compared to expression data obtained by qRT-PCR. Now none of this information is very novel, nor is it of much interest to readers, since no experiments have been undertaken that show evidence for divergent regulatory principles of carotenoid formation as a result of control mechanisms in transcription. The question remains not only about the novelty value of this work or about the added values of this publication. At its best, the manuscript basically contains some redundant information, which have been previously published. The presentation of the results is appealing and convincing. However, this says nothing about the quality and significance of the data presented. The methods used to present the results are well known to the authors. I this context, what remains unclear is, among other things, what the PCA analysis in Figure 3b has yielded in terms of findings.  Furthermore, the labels of the graphs in Fig. 3b are not readable.

Now, some interesting questions could be addressed experimentally. Unfortunately, this has not been done. 1) Why is there no longer a correlation between chlorophyll content and the content of carotenoid species. This correlation is normally observed and actually due to pigment-binding proteins. So if this is no longer true, how are carotenoids assembled in the membrane or by what proteins are they bound?  2.What could argue for a correlation of individual regulated genes and carotenoid or chlorophyll content? Is not post-translational regulation of carotenoid synthesis proteins much more important than regulation of gene expression? 3) What accounts for the developmental correlation of carotenoid content to leaf development? Unfortunately, the regulatory mechanisms that are actually of main interest (line 446) have not been investigated. Thus, the discussion also remains below its potential and is characterized more by generalities and trivial statements.

There are a few grammatical mistakes and a few confusing statements, which deserve text-editing.

Author Response

(The authors gave the same response as above.)

Reviewer 3 Report

I just want to bring up a quick question to the authors: it is well-known that lettuce produces lactucaxanthin, the epsilon, epsilon-carotene-3,3'-diol that is specifically produced by Lactuca sativa, and very limited other species. However, I don't see this compound from the analysis, not the epsilon, epsilon-branch of the metabolic pathway. I don't think this part can be missed or ignored. Please check and provide the HPLC detection and pigment confirmation results in the manuscript before I can make my assessment of this manuscript.

Author Response

(The authors gave the same response as above.)

Round 2

Reviewer 3 Report

With the addition of the information regarding the epsilon,epsilon-branch, I think now the manuscript is better.

The writing is ok.